# 20 kDa PEGylated Adrenomedullin as a New Therapeutic Candidate for Inflammatory Bowel Disease

**Goro Miki [1],[\*]** , **Nobuko Kuroishi [1]**, **Mariko Tokashiki [1]**, **Sayaka Nagata [1]**, **Masaji Tamura [2]**, **Taku Yoshiya [2]** , **Kumiko Yoshizawa-Kumagaye [2]**, **Shinya Ashizuka [1]** , **Joji Kato [3]**, **Motoo Yamasaki [1]** and **Kazuo Kitamura [1]**

[1]  Division of Circulation and Body Fluid Regulation, Faculty of Medicine, University of Miyazaki, 5200, Kibana, Kiyotakecho, Miyazaki 889-1692, Japan; nobuko_kuroishi@med.miyazaki-u.ac.jp (N.K.); mariko_tokashiki@med.miyazaki-u.ac.jp (M.T.); sayaka_nagata@med.miyazaki-u.ac.jp (S.N.); shinya_ashizuka@med.miyazaki-u.ac.jp (S.A.); motoo_yamasaki@med.miyazaki-u.ac.jp (M.Y.); kazuokit@med.miyazaki-u.ac.jp (K.K.)

[2]  Peptide Institute, Inc., 7-2-9 Saito-Asagi, Ibaraki-shi, Osaka 567-0085, Japan; tamura@peptide.co.jp (M.T.); t.yoshiya@peptide.co.jp (T.Y.); kumiko@peptide.co.jp (K.Y.-K.)

[3]  Frontier Science Research Center, University of Miyazaki, 5200, Kibana, Kiyotakecho, Miyazaki 889-1692, Japan; jkjpn@med.miyazaki-u.ac.jp

[\*]  Correspondence: goro_miki@med.miyazaki-u.ac.jp; Tel.: +81-0985-85-9227

**Abstract:** Human adrenomedullin (AM), a hypotensive peptide, also has anti-colitis activity. We prepared a polyethylene glycol (PEG) ylated form of AM through the conjugation of PEG-AM (1–15) and AM (15–52). Highly pure monomeric 20 kDa PEG-AM (20kPEG-AM) stimulated cyclic adenosine monophosphate production in HEK-293 cells stably expressing the type 1 AM receptor in a dose-dependent manner. The half-life of 20kPEG-AM was 7.4 h following subcutaneous administration in mice. We assessed the anti-colitis effect of subcutaneous 20kPEG-AM administration in the dextran sodium sulfate murine colitis model. Single and double subcutaneous injection of 20kPEG-AM significantly reduced total inflammation scores. These results suggest that 20kPEG-AM is a promising therapeutic candidate for the treatment of human inflammatory bowel diseases.

**Keywords:** adrenomedullin; polyethylene glycol; inflammatory bowel disease; colitis; dextran sulfate

## 1. Introduction

Human adrenomedullin (AM), a 52-residue peptide with an amidated C-terminus and an intramolecular disulfide bond, was initially identified as a vasodilatory peptide derived from a human pheochromocytoma [1]. AM is expressed in many organs and tissues, including the gastrointestinal tract, vascular endothelium, adipose tissue, blood, heart, lung, kidney, pancreas, and the adrenal medulla [2,3]. The peptide has multiple functions in anti-inflammation, vasodilation, angiogenesis, wound healing, and protection of the heart, blood vessels, and brain [4,5]. AM showed therapeutic effects in experimental models of ischemic heart disease, stroke, retinochoroidal disease, and insulin intolerance [4,6–9].

AM also showed therapeutic effects in an experimental model of dextran sulfate sodium (DSS)-induced colitis. Continuous or daily subcutaneous injection of AM reduced colitis severity [7]. Moreover, AM showed a therapeutic effect in patients with ulcerative colitis and Crohn's disease following continuous intravenous injection [10,11]. However, these preclinical and clinical applications of AM required continuous or frequent administration because the plasma half-life of AM is only

4.87 ± 0.68 min (mean ± standard error of the mean, S.E.M.) [12]. The N-terminus of AM was conjugated to 60 kDa polyethylene glycol (PEG) to elongate its plasma half-life (60kPEG-AM). It was found that 60kPEG-AM reduced the severity of DSS-induced colitis following a single subcutaneous injection in mice [13]. While 60kPEG-AM represents a potential peptide therapeutic, clinically approved biopharmaceuticals contain PEG polymers only up to 50 kDa in size [14]. With the goal of developing AM therapeutics for clinical applications, we prepared a 20 kDa PEGylated form of AM (20kPEG-AM) and evaluated its physiological, pharmacokinetic, and anti-colitis properties. We also evaluated the anti-colitis effect of repetitive administration of PEGylated AM to simulate the anticipated dosing of human patients in a clinical setting.

## 2. Results

### 2.1. Synthesis and Characterization of 20kPEG-AM

We synthesized 20kPEG-AM by native chemical ligation [15] of the following two segments: (i) the N-terminal segment (PEG20000-AM(1–15)-SCH$_2$CH$_2$CO-Arg-Arg-Arg-NH$_2$) and (ii) the C-terminal segment (AM(16–52)). The N-terminal segment without a PEG moiety was first synthesized by Boc solid-phase peptide synthesis and purified by high-performance liquid chromatography (HPLC). N-terminal PEGylation was performed using PEG20000-NHS (SUNBRIGHT ME-200-HS, NOF Corporation, Tokyo, Japan) at room temperature for 100 min, followed by HPLC purification. The C-terminal segment was synthesized using Boc chemistry in solution according to a previously described method [16] and purified by HPLC. The N- and C-terminal segments were ligated in the presence of thiophenol to yield reduced PEG20000-AM, which was then oxidized with iodine and purified by HPLC to yield 20kPEG-AM. The yield of the ligation reaction was approximately 80%, and amino acid analysis following acid hydrolysis indicated that 20kPEG-AM contained 52 amino acid residues (Table 1). The purity of 20kPEG-AM was determined to be 99% by reverse phase (RP)-HPLC (Figure 1A). Gel filtration chromatography indicated that 20kPEG-AM eluted at the anticipated volume, suggesting that 20kPEG-AM exists as a monomer in solution (Figure 1B).

**Table 1.** Amino acid analysis of 20kPEG-AM.

| Amino Acid (Expected Residues/Molecule) Residues/Molecule | | | |
|---|---|---|---|
| Asp (6) 6.00 | Thr (3) 2.92 | Ser (4) 3.63 | Glu (6) 5.99 |
| Gly (4) 3.96 | Ala (2) 1.99 | Cys (2) 1.85 | Val (2) 2.01 |
| Met (1) 0.93 | Ile (2) 1.96 | Leu (2) 2.00 | Tyr (3) 2.84 |
| Phe (4) 3.97 | Lys (4) 3.98 | NH$_3$ (10) 10.72 | |
| His (1) 1.03 | Arg (4) 4.01 | Pro (2) 2.03 | |

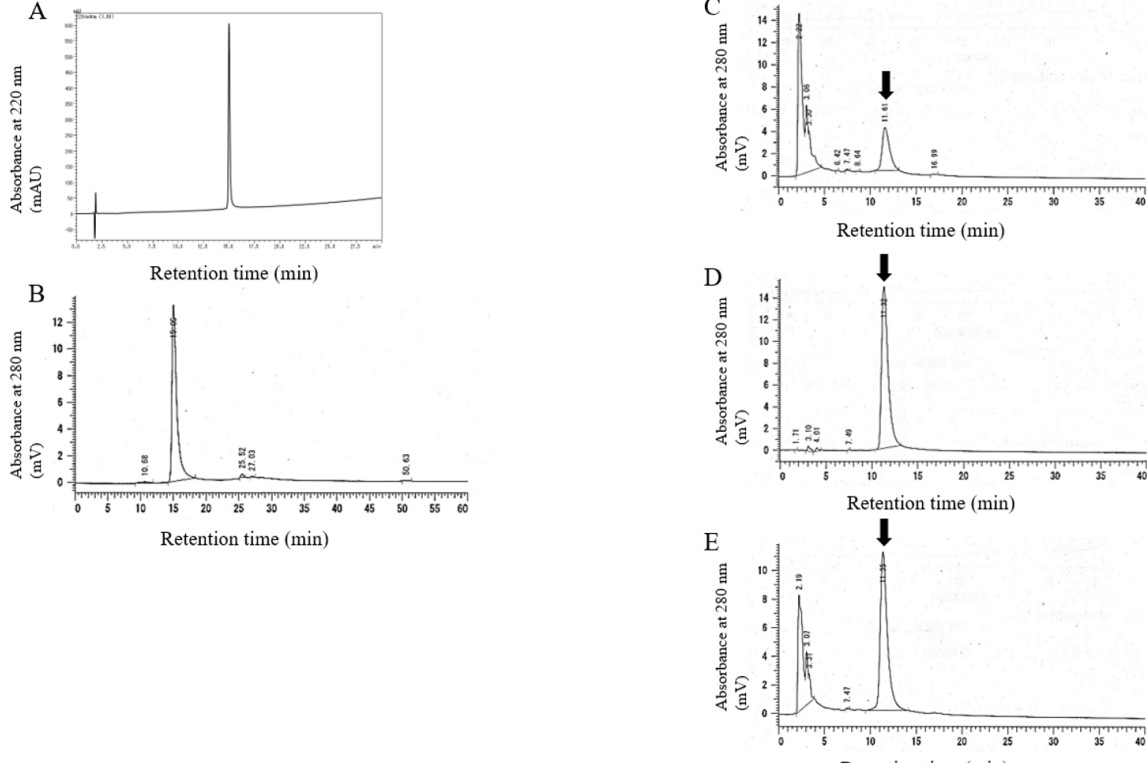

**Figure 1.** Analysis of 20kPEG-AM. Purity was estimated by RP-HPLC (**A**). Analysis of molecular size in a solution using gel filtration chromatography (**B**). Elution profiles from ion-exchange chromatography ((**C**), CNBr-digested 20kPEG-AM; (**D**), AM(6–52); (**E**), coinjection of CNBr-digested 20kPEG-AM and AM(6–52)).

We confirmed that PEG-AM (1–15) and AM (15–52) were successfully conjugated using CNBr digestion [17]. Because AM only contains one Met residue at position 5, CNBr digestion yielded AM (6–52). Figure 1C–E show ion-exchange chromatography of CNBr-digested 20kPEG-AM, AM (6–52), and the coinjection of both molecules. CNBr-digested 20kPEG-AM and AM (6–52) had identical retention times of 11 min, and coinjection yielded a single peak at 11 min, suggesting that 20 kDa PEG had been successfully conjugated to the N-terminus of AM.

## 2.2. Cell Culture Experiments

Native AM and 20kPEG-AM stimulated intracellular accumulation of cyclic adenosine monophosphate (cAMP) in HEK-293 cells (Figure 2). Native human AM elevated intracellular cAMP levels in a dose-dependent manner with a pEC50 of $8.05 \pm 0.09$ (mean $\pm$ S.E.M.) and an Emax of $86.93 \pm 3.91$ nmol/well. 20kPEG-AM exerted similar effects, augmenting cAMP production with a pEC50 of $7.27 \pm 0.08$ and an Emax of $90.80 \pm 2.97$ nmol/well. The pEC50 of 20kPEG-AM was lower than that of native AM, while the Emax values of the two molecules were similar.

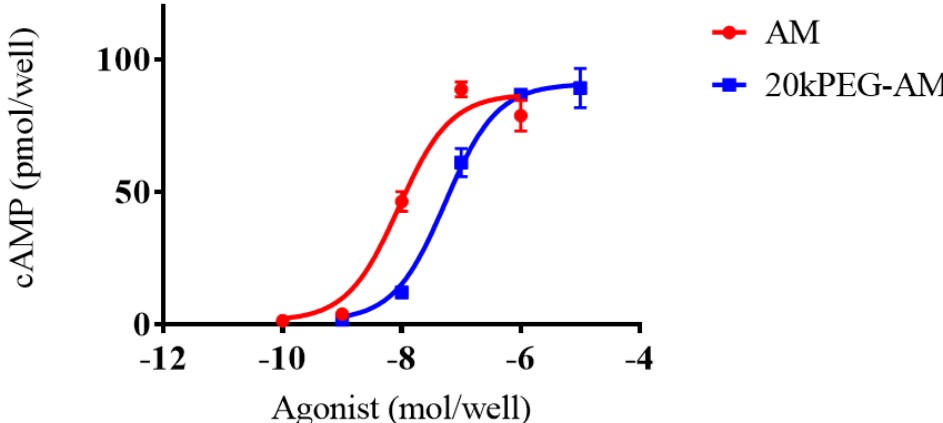

**Figure 2.** Intracellular cAMP accumulation following treatment of cultured HEK-293 cells with 20kPEG-AM or native AM. HEK-293 cells stably expressing the type 1 AM receptor were treated with the indicated concentrations of 20kPEG-AM or native AM for 15 min. Intracellular cAMP levels were measured using an enzyme immunoassay. Data are presented as the means ± S.E.M. of five replicates for each concentration.

*2.3. Plasma Levels of 20kPEG-AM Following Subcutaneous Administration in Mice*

Concentrations of 20kPEG-AM in blood at various time points following subcutaneous administration of 10 nmol/kg 20kPEG-AM in mice are shown in Figure 3. Plasma concentrations were 18.73 ± 1.57 pM (means ± S.E.M.) on day 4 and 16.23 ± 0.69 pM on day 8. The first and second plasma half-life of 20kPEG-AM was 7.40 h (95% confidence interval: 4.4–12.9 h). This represented a more than 100-fold extended half-life for 20kPEG-AM compared with native AM [12]. No toxicity or adverse effects were noted in treated mice.

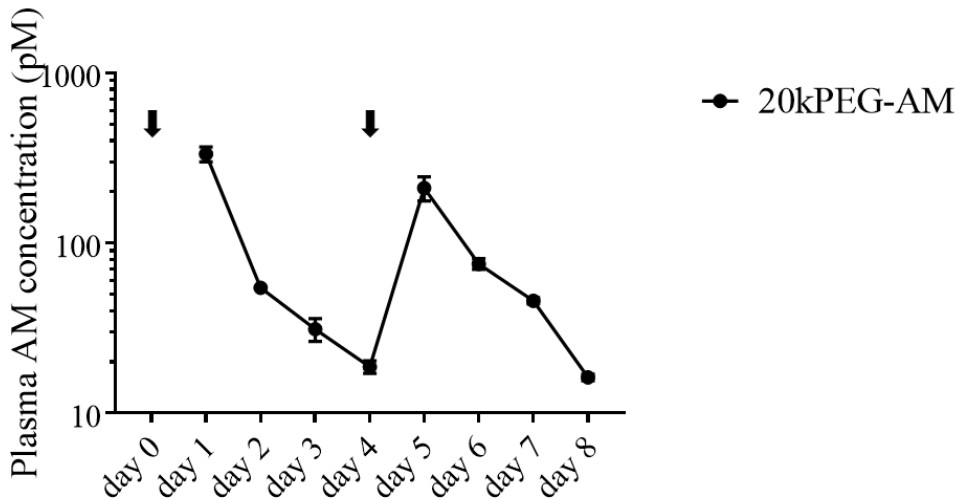

**Figure 3.** Plasma clearance of 20kPEG-AM. 20kPEG-AM (10 nmol/kg) was administered by subcutaneous injection on days 0 and 4. Blood samples were collected at the indicated time points. Data are presented as means ± S.E.M. Three mice were examined at each time point.

*2.4. Induction of DSS Colitis and Treatment with 20kPEG-AM*

Induction of DSS colitis and treatment with 20kPEG-AM was performed, as shown in Figure 4. The mortality of this experiment was zero.

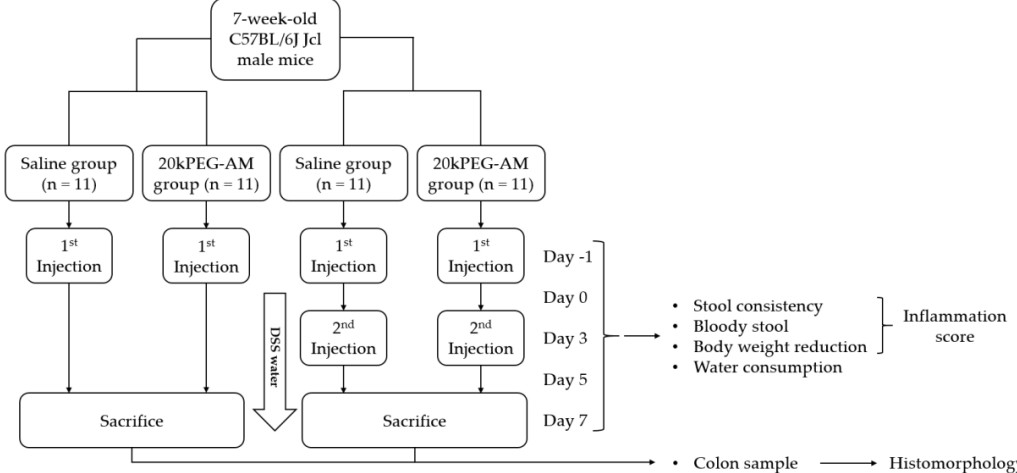

**Figure 4.** Flow chart of an animal experiment that evaluated an anti-colitis effect of 20kPEG-AM. Single and double injection protocols were performed with saline as a control. DSS water was given *ad libitum* from days 0 to 7. Mice were checked for stool consistency, bloody stool, bodyweight rection, and water consumption on days −1, 0, 3, 5, and 7.

Following a single injection of 20kPEG-AM, there were trends toward lower stool consistency scores (Figure 5A), bloody stool scores (Figure 5B), and bodyweight reduction scores (Figure 5C) in the 20kPEG-AM group at day 7, but these differences did not reach statistical significance. The total inflammation scores were significantly reduced at day 7 ($p < 0.001$, Figure 5D). There was no statistical difference in DSS water consumption between the two groups (data not shown), implying that 20kPEG-AM did not ameliorate colitis by decreasing the DSS water consumption.

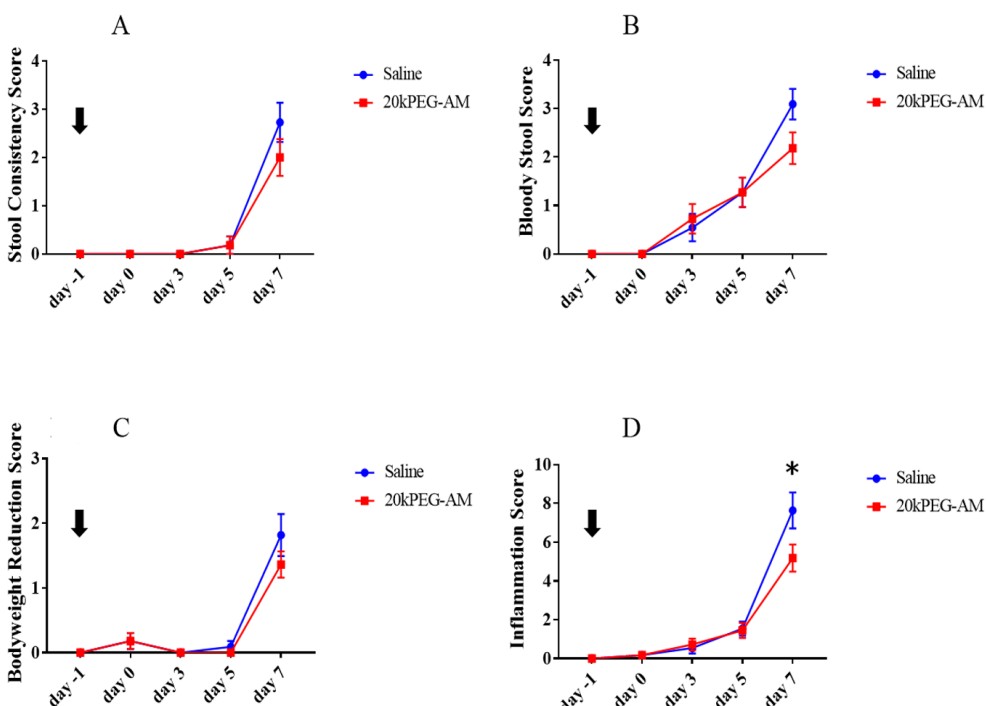

**Figure 5.** Comparison of scores in the DSS-induced colitis model following treatment with 20kPEG-AM (n = 11 mice/group). Single-injection protocol: (**A**) stool consistency score, (**B**) bloody stool score, (**C**) bodyweight reduction score, and (**D**) inflammation score. Results are shown as means ± S.E.M. * $p < 0.001$ versus control.

In the double injection protocol, stool consistency scores ($p < 0.001$, Figure 6A), bodyweight reduction scores ($p < 0.001$, Figure 6B), and total inflammation scores ($p < 0.001$, Figure 6C) were significantly lower in the 20kPEG-AM group at day 7. There was a trend toward lower bloody stool scores (Figure 6D) in the 20kPEG-AM group at day 7, but this difference did not reach statistical significance. Histology of control group mice (Figure 7A) showed more severe colitis compared with 20kPEG-AM-treated mice (Figure 7B; single and double injection protocols). However, the scoring of the histomorphological evaluation showed no statistically significant differences (data not shown).

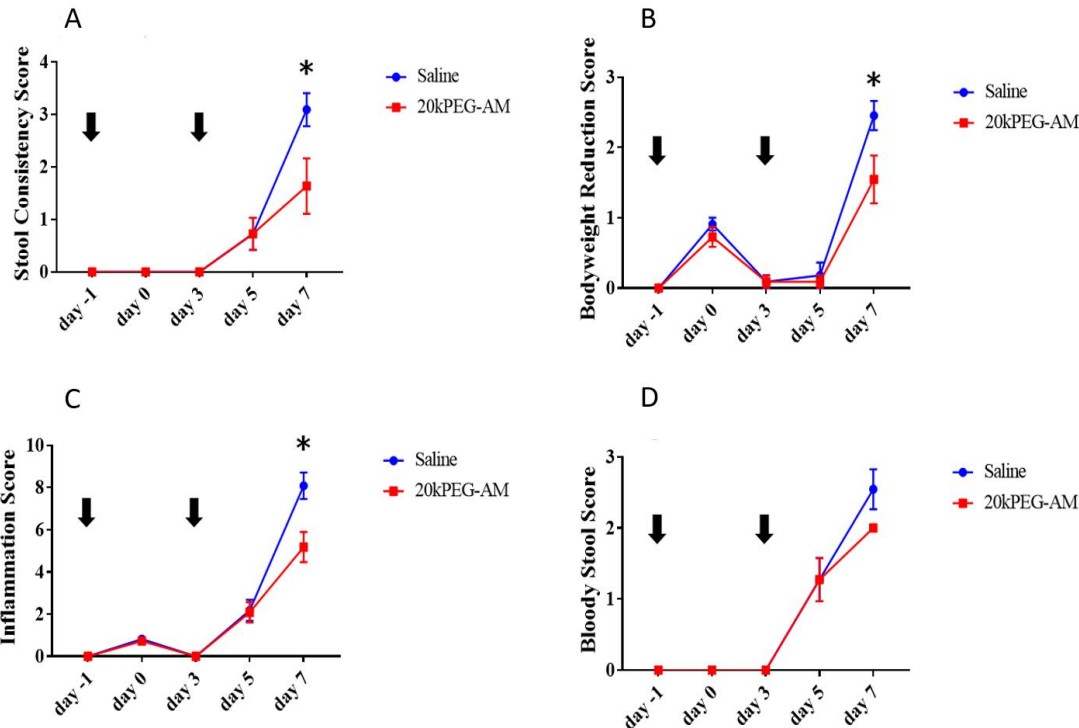

**Figure 6.** Comparison of scores in the DSS-induced colitis model following treatment with 20kPEG-AM (n = 11 mice/group). Double injection protocol: (**A**) stool consistency score, (**B**) bodyweight reduction score, (**C**) inflammation score, and (**D**) bloody stool score. Results are shown as means ± S.E.M. * $p < 0.001$ versus control.

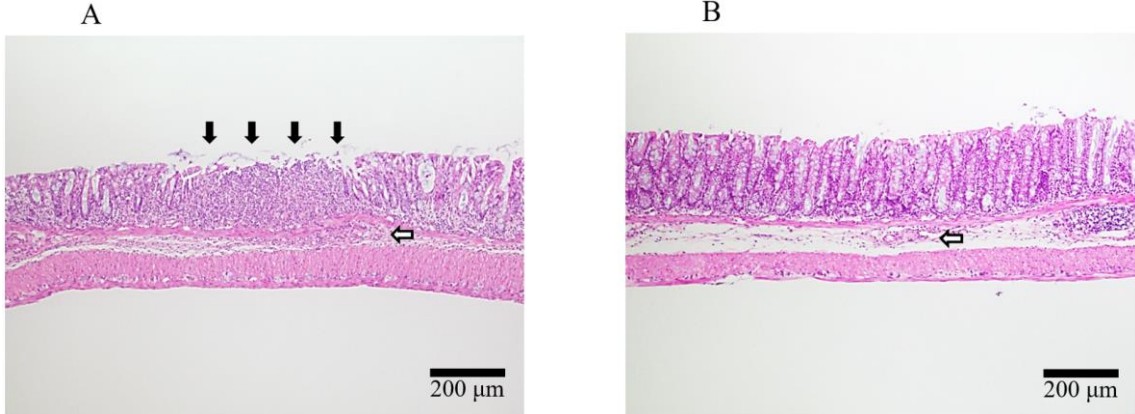

**Figure 7.** Histology of control group mice (**A**) and 20kPEG-AM-treated mice (**B**). Figure 7A shows focal ulceration (black arrow) with mucosal and submucosal immune cell infiltration (white arrow). Figure 7B shows no architectural change with mild inflammatory cell infiltrates in the submucosa (white arrow). The scoring of the histomorphological evaluation showed no statistically significant differences.

## 3. Discussion

Although many new drugs for inflammatory bowel diseases have been developed, there are notable cases of treatment resistance, and the development of new drugs is urgently needed. In this study, we developed 20kPEG-AM and demonstrated its anti-colitis effect following both single and double subcutaneous injection in mice. Our findings suggest that 20kPEG-AM represents a new candidate therapeutic for inflammatory bowel disease.

PEGylated biologics have pharmacological advantages over the unmodified parental molecules, including extended circulating half-life, reduced dosing frequency, lower toxicity, decreased kidney clearance, decreased immunogenicity, maintenance of biological activity, increased stability, and protection from proteolytic degradation [18–20]. We sought to use PEGylation to overcome the short plasma half-life of AM because the extended half-life of 20kPEG-AM would make continuous intravenous injection unnecessary in a clinical setting.

We successfully prepared 20kPEG-AM by native chemical ligation of an N-terminal (PEG20000-AM(1–15)-SCH$_2$CH$_2$CO-Arg-Arg-Arg-NH$_2$) and a C-terminal segment (AM(16–52))20 kDa PEG [15]. Successful conjugation was confirmed using HPLC following CNBr digestion and amino acid analysis. The efficiency of ligation between the N- and C- terminal segments was sufficient to enable the industrial-scale synthesis of 20kPEG-AM. In addition, purity was high enough for clinical applications. 20kPEG-AM increased cAMP production in treated HEK-293 cells with a pEC50 value of 7.27 ± 0.08, slightly lower than that of native AM. However, the plasma half-life of 20kPEG-AM was 7.4 h (95% confidence interval: 4.4–12.9 h), an extension of more than 100-fold compared with native AM [12].

An anti-colitis effect was exerted following the subcutaneous injection of 20kPEG-AM in a mouse model of DSS-induced colitis. Our findings are consistent with those of previous studies showing that PEGylated AM has an anti-colitis effect [13]. However, the PEG used in our study was smaller than 50 kDa. The maximum molecular weight of PEGs used in currently approved PEGylated biotherapeutics [14]. As the approval process for new drugs with very different molecular sizes can be challenging, our data offer a clear path for clinical development of PEGylated AM as a biotherapeutic for inflammatory bowel diseases.

Repetitive subcutaneous injection of 20kPEG-AM exerted an anti-colitis effect in mice. For the treatment of chronic diseases such as inflammatory bowel disease, repetitive administration of PEGylated peptide biotherapeutics is typically required [21]. Our report is the first to demonstrate that repetitive administration of PEGylated AM has an anti-colitis effect. A limitation of this finding is that a direct comparison between single and double injection protocols was not performed. Nonetheless, our data provide a foundation for future studies investigating the long-term anti-colitis effect of 20kPEG-AM.

Our study provides evidence on potential target trough plasma AM concentrations capable of exerting anti-colitis effects following administration of PEGylated AM. In this study, 20kPEG-AM exerted an anti-colitis effect when the trough AM concentrations were 18.73 ± 1.57 pM and 16.23 ± 0.69 pM on days 4 and 8, respectively. However, a previous study of 60kPEG-AM showed that much higher trough AM concentrations were required to exert an anti-colitis effect. Doses of 5 and 25 nmol/kg showed an anti-colitis effect, and a dose of 10 nmol/kg yielded a trough AM concentration of 865 pM on day 7. In clinical applications, the frequency of administration and dose of PEGylated AM are designed to achieve the targeted trough AM concentration while avoiding potential toxicity from overdosing. Our data serve as an essential foundation for future clinical efficacy and safety studies of PEGylated AM.

We found that 20kPEG-AM exerted anti-colitis effects at considerably lower doses, in terms of the human equivalent dose (HED) [22], compared with two studies of native AM administration. While native AM required the administration of approximately 43 nmol/kg every day in humans [10] and 0.65–6.5 nmol/kg HED every day in mice to achieve an anti-colitis effect [7], 20kPEG-AM only required a dose of 0.81 nmol/kg HED administered every 4 days. This result was unexpected because our study also showed that 20kPEG-AM had relatively low in vitro efficacy compared with native AM.

Other than the difference of administration route, one explanation of this result may be that PEGylated molecules can have improved functional pharmacodynamic properties for their natural receptors despite low efficacy in vitro [23,24]. This improvement may also explain the fact that a single injection was sufficient to exert an anti-colitis effect, despite relatively low plasma AM concentrations on day 4. The potential to minimize dosing is an advantage for clinical applications because reducing the dosage is likely to minimize side effects, including hypotension [25,26].

The mechanism underlying the anti-colitis effect of AM has yet to be discovered. Many studies have shown that increased cAMP concentrations can improve epithelial barrier function [27–29]. It was recently reported that intravenous administration of AM resulted in increased cAMP levels in the colonic mucosa of mice showing ameliorated DSS-induced colitis [7]. These findings imply that AM-induced cAMP in the mucosa may contribute to epithelial barrier protection and anti-colitis effects, downregulating proinflammatory cytokines such as phosphorylated NF-kB, tumor necrosis factor-$\alpha$, interleukin (IL)-1$\beta$, IL-6, interferon-$\gamma$, IL-4, IL-12, and keratinocyte chemoattractant [7,30–32]. Another mechanism could involve the upregulation of epithelium regeneration markers, such as Lgr5, Wnt5a, Egfr, or Erbb2 [33], and regulation of gut bacterial composition through downregulation of TLR4 [34]. Additional studies are needed to define the molecular mechanism of AM's anti-colitis action.

One of the limitations of our study was the potential for confirmation bias. Although we observed a statistically significant difference in mice treated with 20kPEG-AM in one objective measure (bodyweight reduction score), other measures such as stool consistency and bloody stool scores are more subjective and susceptible to bias.

## 4. Materials and Methods

### 4.1. Analysis of 20kPEG-AM

The elution profile of 20kPEG-AM was assessed by gel filtration chromatography using a Superdex 200 Increase 10/300 GL column (GE Healthcare UK, Little Chalfont, England) with 20 mM sodium citrate containing 100 mM NaCl (pH 7.0) as the mobile phase. The purity of 20kPEG-AM was estimated by RP-HPLC with a Zorbax 300SB-C18 4.6 × 150 mm column (Agilent Technologies, Inc., Santa Clara, CA, USA) using gradient elution (20–70% CH$_3$CN/0.1% trifluoroacetic acid). The PEGylation site of 20kPEG-AM was confirmed by ion-exchange chromatography with a CM-2SW 0.46 × 25 cm column (Tosho, Tokyo, Japan) following CNBr digestion as described previously [13]. Amino acid analysis following acid hydrolysis was conducted, as shown in Table 1.

### 4.2. Cell Culture Experiments

To assess the physiological properties of PEGylated AM in vitro, HEK-293 cells stably expressing the type I AM receptor were treated with native AM or 20kPEG-AM. Intracellular cAMP accumulation was monitored as described previously [12]. Cells were treated with peptide concentrations ranging from $10^{-10}$ M to $10^{-6}$ M for native AM and from $10^{-9}$ M to $10^{-5}$ M for 20kPEG-AM.

### 4.3. Animal Experiments

4.3.1. Plasma Concentration of 20kPEG-AM Following Subcutaneous Administration in Mice

To assess the pharmacokinetic properties of 20kPEG-AM, male 7-week-old C57BL/6J Jcl mice were purchased from CLEA Japan, Inc. (Tokyo, Japan) and housed under specific pathogen-free conditions with a 12-h light/12-h dark cycle and a standard diet. The study was performed following the guidelines of the Animal Welfare Act and with the approval of the University of Miyazaki Institutional Animal Care and Use Committee (2012-528, 31 Mar. 2016).

On days 0 and 4, 20kPEG-AM (10 nmol/kg) was subcutaneously administered. Blood samples were obtained under 3% isoflurane anesthesia (Isoflu, Zoetis, Tokyo, Japan). The abdominal cavity was opened, and blood drawn from the inferior vena cava was transferred to ice-cold tubes containing

aprotinin (500 KIU/mL) and ethylenediaminetetraacetic acid (1 g/L). Thereafter, the animals were sacrificed by exsanguination. Three mice per day were sacrificed from days 1 to 8. After centrifugation at $1700 \times g$, 20kPEG-AM levels in plasma were measured using a fluorescence enzyme immunoassay [12]. The data were displayed as a semilogarithmic plot and fitted to the equation for one phase exponential decay using Prism version 7.02.

### 4.3.2. Induction of DSS Colitis and Treatment with 20kPEG-AM

To assess the pharmacological properties of 20kPEG-AM, male 7-week-old C57BL/6J Jcl mice were purchased from CLEA Japan, Inc. and housed under specific pathogen-free conditions with a 12-h light/12-h dark cycle and a CRF-1 (Oriental Yeast Co., Ltd, Tokyo, Japan) diet. The study was performed following the guidelines of the Animal Welfare Act and with the approval of the University of Miyazaki Institutional Animal Care and Use Committee (2012-528, 31 Mar. 2016).

For the single injection protocol, 20kPEG-AM (10 nmol/kg, $n = 11$) or saline (control, $n = 11$) were subcutaneously administered to mice on day −1. Thereafter, water containing DSS (3% w/v, 5000 kDa, Wako, Osaka, Japan) was given ad libitum from days 0 to 7. For the double injection protocol, 20kPEG-AM (10 nmol/kg, $n = 11$) or saline (control, $n = 11$) were subcutaneously administered to mice on days −1 and 3. DSS water was given as above.

Mice were assessed for stool consistency, bloody stool, and bodyweight reduction, as shown in Table 2 on days −1, 0, 3, 5, and 7. The total inflammation score was calculated by combining these scores [35]. Mice were also checked for the water consumption on the days mentioned above. The bodyweight of mice was measured by individually placing mice onto an electronic balance. The percent body weight loss was calculated by dividing the daily bodyweight by the weight at day -1. We exchanged the cage without bedding on days −1, 0, 3, 5, and 7 to obtain freshly excreted feces. Only the feces with moisture but not in contact with urine were observed. On day 7, colons were collected by abdominal operation under 3% isoflurane anesthesia (Isoflu, Zoetis, Tokyo, Japan). The animals were sacrificed by cervical dislocation. A total of 44 colon samples were obtained for evaluation from single and double injection protocols. Colons were fixed with 10% formaldehyde neutral buffer solution (Nacalai Tesque Inc., Kyoto, Japan). They were then embedded in paraffin and stained with hematoxylin and eosin. Tissues were single blindly scored using the histomorphological evaluation of chemically induced colonic inflammation in mouse models, as shown in Table 3 [36].

**Table 2.** Assessment of diarrhea, bloody stool, and bodyweight reduction in mice with DSS-induced colitis for calculation of total inflammation score.

| Score | Stool Consistency | Bloody Stool | Bodyweight Reduction |
|:---:|:---:|:---:|:---:|
| 0 | Normal | No blood | No reduction |
| 1 | | | 1–5% reduction |
| 2 | Soft | Fecal occult blood | 5–10% reduction |
| 3 | | | 10–20% reduction |
| 4 | Diarrhea | Mucus and bloody stool | |

**Table 3.** Scoring of histomorphology for chemically-induced colonic inflammation.

| Inflammatory Cell Infiltrate | | | Intestinal Architecture | | |
|:---:|:---:|:---:|:---:|:---:|:---:|
| Severity | Extent | Score 1 | Epithelial Changes | Mucosal Architecture | Score 2 |
| Mild | Mucosa | 1 | Focal erosions | | 1 |
| Moderate | Mucosa and submucosa | 2 | Erosions | ± Focal ulcerations | 2 |
| Marked | Transmural | 3 | | Extended ulcerations ± granulation tissue ± pseudopolyps | 3 |
| | | | Sum of scores 1 and 2: | | 0–6 |

All data were analyzed using GraphPad Prism version 7.02 (GraphPad, La Jolla, CA, USA). Differences between two groups were assessed using the unpaired t-test, and differences among multiple groups were assessed using analysis of variance followed by the Sidak post-hoc test. All data were expressed as means ± S.E.M. unless otherwise indicated. Values of $p < 0.05$ were considered statistically significant.

## 5. Conclusions

Our findings provide a foundation for further research towards preclinical and clinical applications of 20kPEG-AM for human inflammatory bowel disease.

**Author Contributions:** Conceptualization, methodology, formal analysis, investigation, writing—original draft, visualization, G.M.; investigation, M.T. (Mariko Tokashiki), N.K.; methodology, investigation, writing—review and editing, S.N.; methodology, investigation, resources, writing—review and editing, M.T. (Masaji Tamura), T.Y., and K.Y.-K.; methodology and funding acquisition, S.A.; conceptualization, methodology, writing—review and editing, supervision, project administration, and funding acquisition, J.K., M.Y., and K.K. All authors have read and agreed to the published version of the manuscript.

**Funding:** This work was partly supported by Grants-in-Aid for Scientific Research from the Japan Society for the Promotion of Science (16K09316 and 18H02810) and a Scientific Research Grant for Creating Start-ups from the Advanced Research and Technology (START) Program of the Japan Science and Technology Agency (ST262010WV).

**Acknowledgments:** We thank Koji Negishi for excellent technical assistance.

**Conflicts of Interest:** K. Kitamura and M. Yamasaki have stocks in Himuka AM Pharma Corporation.

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
