# Peer review of "20 kDa PEGylated Adrenomedullin as a New Therapeutic Candidate for Inflammatory Bowel Disease"

_gastrointestdisord, doi:10.3390/gidisord2040033_

Round 1

Reviewer 1 Report

I appreciate the opportunity to review this interesting paper on the anti-colitis effect of repetitive administration of PEGylated Adrenomedullin (AM).

The genesis of this paper is the observation that AM also showed therapeutic effects in an experimental model of dextran sulfate sodium (DSS)- induced colitis and also showed a therapeutic effect in IBD patients. However, AM has to be continuous or frequently administered since the plasma half-life of AM is very short. The authors prepared a 20 kDa PEGylated form of AM  (20 kPEG-AM) to extend its plasma half-life and decided to evaluate its physiological, pharmacokinetic and anti-colitis properties.

The authors decided to investigate these properties on two models.

First, to assess the physiological properties of PEGylated AM in vitro, HEK-293 cells stably expressing the type I AM receptor were treated with native AM or 20kPEG-AM, and intracellular cAMP accumulation was monitored.

Second, in male 7-week-old C57BL/6J mice plasma concentration of 20kPEG-AM following subcutaneous administration in mice was determined. Also, in week-old C57BL/6J mice, the protective effects of 20kPEG-AM were studied in mice with DSS- induced colitis.

They observed that native AM and 20 kPEG-AM stimulated intracellular cAMP accumulation in HEK cells. After subcutaneous administration of 20 kPEG-AM, they found a more than 100-fold longer half-life of 20 kPEG-AM compared to native AM. They also showed that repeated subcutaneous injections of 20 kPEG-AM exerted anti-colitis effects in mice.

 The main strengths of this paper are that it addresses an interesting and timely question, the article is well constructed, the experiments were well conducted, and analysis was well performed.

Considering these strengths, though, as I read the manuscript, I found some areas in which I would have appreciated greater clarity. I believe the paper could be further strengthened by added information about:

Methods.

  • A flow chart showing the experimental procedure should be included as it makes the experimental setting more visible to the reader.
  • Induction of DSS colitis should be described with more detail.
  • Did water intake was observed in each group of mice by measuring leftover water? This measurement is essential because some after administration of some substances mice may consume more DSS water than their controls, thereby developing relatively more severe colitis.
  • Please describe in more detail how the measurement of body weight and the presence of occult blood was made. How faeces were collected and how occult blood was monitored?
  • Did you measure DSS-induced colonic mucosal damage in vivo?
  • Gross picture of colons with arrows indicating, for example, superficial inflammation would be helpful.
  • The image of the histological specimens should be more precisely described with arrows indicating, for instance, epithelial erosion or immune cell infiltration in DSS-treated mice. Typical histological changes induced by DSS include mucin and goblet cell depletion, epithelial erosion, and ulceration. Further, DSS induces an influx of neutrophils into the lamina propria and submucosa. Did you observe such a picture?
  • Histologic scoring should be described in detail. Was it blinded?
  • How high mortality was observed in both groups? A relatively high dose of DSS was used.
  • Have you measured proinflammatory biomarkers? If you did not do it, why did you decide so?

Results

  • Due to the structure of the manuscript, the different groups are not appropriately described. For instance, what does NS on figures mean?

Conclusions

Given the inconclusive results, the conclusions do not seem fully justified.

Author Response

Reply to reviewer 1

Methods.

  1. A flow chart showing the experimental procedure should be included as it makes the experimental setting more visible to the reader.

 I have added a new flow chart, which clarifies the experimental procedure. (line 141-147)

  1. Induction of DSS colitis should be described with more detail. Did water intake was observed in each group of mice by measuring leftover water? This measurement is essential because some after administration of some substances mice may consume more DSS water than their controls, thereby developing relatively more severe colitis.

 I have the data on water consumption on days -1, 0, 3, 5, and 7. It was measured by measuring the weight of the bottle with leftover water. I have added sentences in the method (line 311-315) and results (line 146-147,152-154). Also, I briefly addressed what this implies in the method section (line 152-154). I avoided the discussion section, afraid of becoming too busy.

  1. Please describe in more detail how the measurement of body weight and the presence of occult blood was made. How faeces were collected and how occult blood was monitored?

 I have revised line 308 to 312 as "Mice were also checked for the water consumption on the days mentioned above. The bodyweight of mice was measured by individually placing mice onto an electronic balance. The percent body weight loss was calculated by dividing the daily bodyweight by the weight at day -1. We exchanged the cage without bedding on days -1, 0, 3, 5, and 7 to obtain freshly excreted feces. Only the feces with moisture but not in contact with urine were observed." I took time to observe fresh feces since minor and major bleeding will become similarly black if not fresh. The stool consistency also changes dramatically over time.

  1. Did you measure DSS-induced colonic mucosal damage in vivo? Gross picture of colons with arrows indicating, for example, superficial inflammation would be helpful.

 I have observed collected colon sample pictures. However, I was unable to locate erosions and ulcerations because of transparent sub-serosal fat tissue. Also, I did not use the time to pigmentate the colon to avoid mucosal damage of the already damaged colon before fixation by formaldehyde

  1. The image of the histological specimens should be more precisely described with arrows indicating, for instance, epithelial erosion or immune cell infiltration in DSS-treated mice. Typical histological changes induced by DSS include mucin and goblet cell depletion, epithelial erosion, and ulceration. Further, DSS induces an influx of neutrophils into the lamina propria and submucosa. Did you observe such a picture?

I have added a description of the architectural change (ulceration) and immune cell infiltration in the figure 7 caption. Also, indicating arrows in figure 7 were added (line 175-180).

  1. Histologic scoring should be described in detail. Was it blinded?

 The histomorphological evaluation was single-blinded by covering the slide label with tape. The evaluation was made after shuffling and temporarily labeling the slides. The evaluation was supervised by a colleague who had performed DSS colitis mice experiments in the Pathology Department.

  1. How high mortality was observed in both groups? A relatively high dose of DSS was used.

 The mortality of our experiment was zero since our experiment only observed the mice up to day 7. I would also assume that the mortality would have been higher in days 10 and 14. I have added the sentence "The mortality of this experiment was zero." (line 139-140)

  1. Have you measured proinflammatory biomarkers? If you did not do it, why did you decide so?

 I believe that the measurement of inflammatory biomarkers would have added depth to understanding the anti-colitis effect mechanism. However, since our initial goal was to demonstrate an anti-colitis effect of 20kPEG-AM, it was not included in our approved study design.

Results

  1. Due to the structure of the manuscript, the different groups are not appropriately described. For instance, what does NS on figures mean?

 I apologize for the confusion. I have changed "NS" (normal saline) to "saline" in the manuscript. 

Conclusions

  1. Given the inconclusive results, the conclusions do not seem fully justified.

 Firstly, I apologize if I have not understood as the reviewer had intended.

 Taking "inconclusive results" as PEG20k-AM not showing an anti-colitis effect in every evaluated measure, therefore, not justified to state that "an anti-colitis effect was exerted," I have deleted "Single and repetitive administration of 20kPEG-AM showed an anti-colitis effect in a mouse model of inflammatory bowel disease." in the conclusion section and left "Our findings provide a foundation for further research toward preclinical and clinical applications of 20kPEG-AM for human inflammatory bowel disease." (line 349-350)

I want to thank the editor, assistant editor, and the reviewers for taking their precious time to review and process this paper in the current turbulence of COVID-19. I hope that everyone is safe, and the revised manuscript is suitable for publication in gastrointestinaldisorders.

Sincerely,

Goro Miki

Reviewer 2 Report

In the present basic science study Miki et al showed that PEGylated adrenomedullin (AM) stimulated cAMP production in cell cultures and suppressed inflammation in DSS colitis murine model. This is a nice and well planned study. I have only few objections:

1) HEK293 is a kidney-derived cell line. Why did Authors use HEK293 instead of a colonic cell line such as CaCO2?

2) The cell culture model was used only to evaluate cAMP. Authors could have evaluated variations in pro- and anti-inflammatory cytokines as well.

3) Figures 4-5: what is the meaning of NS? Consider to replace it with “control” or “saline solution”.

Author Response

Reply to reviewer 2

  • HEK293 is a kidney-derived cell line. Why did Authors use HEK293 instead of a colonic cell line such as CaCO2?

Using CaCO2 cell line is preferable since it derives from a colonic cell, and our target is the protection of the colon. However, in order to measure the physiological activity of AM, stably expressed AM1 receptor is vital. Therefore, we used the HEK-293 cell line that stably expresses the type I AM receptor, which CaCO2 may lack.

  • The cell culture model was used only to evaluate cAMP. Authors could have evaluated variations in pro- and anti-inflammatory cytokines as well.

The pro- and anti-inflammatory cytokines in the cell culture model are of interest in the effect of 20kPEG-AM. However, the HEK-293 cell line used in this experiment is specially designed for the second messenger evaluation of AM. Therefore, we were not able to evaluate pro- and anti-inflammatory cytokines.

  • Figures 4-5: what is the meaning of NS? Consider to replace it with "control" or "saline solution".

I apologize for the confusion. The word "NS" was replaced by "saline" in the manuscript.

Round 2

Reviewer 1 Report

The authors have addressed satisfactorily most of the points raised by the reviewer.